



# Variable metabolic responses of Skagerrak invertebrates to low $O_2$ and high $CO_2$ scenarios

Aisling Fontanini[1], Alexandra Steckbauer[2], Sam Dupont[3] and Carlos M. Duarte[4]

[1] Department of Environment and Agriculture, Curtin University of Technology, Bentley 6102 WA, Australia
[2] Global Change Research Department, IMEDEA (CSIC-UIB), Instituto Mediterráneo de Estudios Avanzados, C/ Miquel Marqués 21, 07190 Esporles, Spain
[3] Department of Biological and Environmental Sciences, University of Gothenburg, The Sven Lovén Centre for Marine Infrastructure - Kristineberg, 45178 Fiskebäckskil, Sweden
[4] Red Sea Research Center (RSRC), King Abdullah University of Science and Technology (KAUST), Thuwal 23955-6900,
Kingdom of Saudi Arabia

*Correspondence to:* Alexandra Steckbauer (asteckbauer@imedea.uib-csic.es, steckbauer.ocean@gmail.com)

**Abstract**

Coastal hypoxia is a problem that is predicted to increase rapidly in the future. At the same time we are facing rising atmospheric $CO_2$ concentrations, which are increasing the $p$$CO_2$ and acidity of coastal waters. These two drivers are well studied in isolation however; the coupling of low $O_2$ and pH is likely to provide a more significant respiratory challenge for slow moving and sessile invertebrates than is currently predicted. The Gullmar Fjord in Sweden is home to a range of habitats such as sand and mud flats, seagrass beds, exposed and protected shorelines, and rocky bottoms. Moreover, it has a
history of both natural and anthropogenically enhanced hypoxia as well as North Sea upwelling, where salty water reaches the surface towards the end of summer and early autumn. A total of 11 species (Crustacean, Chordate, Echinoderm and Mollusc) of these ecosystems were exposed to four different treatments (high/low oxygen and low/high $CO_2$; varying $p$$CO_2$ of 450 and 1300 ppm and $O_2$ concentrations of 2-3.5 and 9-10 mg $L^{-1}$) and respiration measured after 3 and 6 days, respectively. This allows us to evaluate respiration responses of species of contrasting habitats and life-history strategies to
single and multiple stressors. Results show that the responses of the respiration were highly species specific as we observed both synergetic as well as antagonistic responses, and neither phylum nor habitat explained trends in respiratory responses. Management plans should avoid the generalized assumption that combined stressors will results in multiplicative effects and focus attention on alleviating hypoxia in the region.

**KEYWORDS:** Hypoxia, ocean acidification, low $O_2$, high $CO_2$, low pH, invertebrates, Gullmar Fjord.





## 1    Introduction

Resolving the responses of marine organisms to the multiple pressures associated with global change is a major challenge for marine scientists (Duarte et al., 2014). This challenge is particularly pressing for coastal ecosystems, where human populations and impacts often concentrate. Among these pressures, decreasing $O_2$ concentrations (hypoxia) and
ocean acidification (OA) are receiving particular attention (Diaz and Rosenberg, 2008; Doney et al., 2009; Vaquer-Sunyer and Duarte, 2008; Kroeker et al., 2013). Whereas uptake of anthropogenic $CO_2$, is leading to more acid waters in the open ocean (Doney et al., 2009; Caldeira and Wickett, 2003), the control of pH in coastal waters is more complex, often involving metabolic processes (Duarte et al., 2013). This metabolic control is particularly evident when eutrophication stimulates algal blooms, leading to increased sedimentation of organic matter, subsequently degraded through microbial respiration, thereby
consuming $O_2$ and releasing $CO_2$ (Conley et al., 2009). Hence, elevated $CO_2$ through excess respiration is associated with reduced $O_2$ in coastal ecosystems, as these two gases are linked through metabolic processes. Indeed hypoxia is affecting a growing number of coastal ecosystems (Diaz and Rosenberg, 2008; Vaquer-Sunyer and Duarte, 2008), suggesting that eutrophication-driven acidification (Borges and Gypens, 2010; Cai et al., 2010) should be spreading as well. Whereas scientists have acknowledged this coupling over decades, the impacts of these two pressures have continued to be studied in
isolation, although the combined stress from depleted $O_2$ and high $CO_2$ is likely to provide a significant challenge to coastal invertebrates and less mobile species.

The consequences of high $CO_2$ for marine organisms reach further than the highly documented impacts on calcification rates (Doney et al., 2009). For example, the regulation of internal acid-base balance is a challenge as some organisms are unable to compensate for increased environmental acidity (e.g. Pane and Barry, 2007), which may lead to
elevated $CO_2$ and low pH in their blood fluids, referred to as hypercapnia, reducing the affinity of haemoglobin for $O_2$ and further interfering with respiratory processes. Depending on the severity of the pH change, organisms can experience mortality and a range of sub-lethal effects such as increased respiration, reduced growth, behavioural changes such as compromised ability to avoid predators (see summary by Kroeker et al., 2013), and increased susceptibility to parasites (Burgents et al., 2005). Similarly, hypoxia has been shown to cause mortality, reduced growth and reproduction, decrease
respiration rates, and induce behavioural changes such as forced migrations, which may make organisms more susceptible to predation (Vaquer-Sunyer and Duarte, 2008). There are growing concerns that the combined impacts of elevated $CO_2$ and hypoxia may prove to be a more significant challenge for marine life that the predictions from isolated effects (Burnett, 1997; Brewer and Peltzer, 2009; Mayol et al., 2012; Melzner et al., 2013). While other studies have considered combined stressors such as low $O_2$ and high/low temperature (reviewed by Vaquer-Sunyer and Duarte, 2011), low $O_2$ and increased
hydrogen sulphide (reviewed by Vaquer-Sunyer and Duarte, 2010), increased acidity and temperature (Doney et al., 2009; Lischka et al., 2010), research focusing on how high $p$CO$_2$ and low $O_2$ will impact marine organisms has been of recent attention (e.g. Gobler et al., 2014; Steckbauer et al., 2015; Sui et al., 2016ab). For example, recent reports have shown that low $O_2$ and high $p$CO$_2$ can cause additively impacts for the growth and survivorship of bivalve larvae and later stage clams (Gobler et al., 2014), however similar research indicates some early life stage bivalves are largely tolerant of these combined
effects (Frieder et al., 2014). Moreover, invertebrates along the coastline of Chile show rather additive than synergetic affects to the combination of low $O_2$ and high $CO_2$ (Steckbauer et al., 2015).

Here we examine the independent and joint impacts of high $p$CO$_2$ and low $O_2$ scenarios on the respiration of Skagerrak marine invertebrates at projected future levels. We did so by examining the responses of 11 species of Skagerrak marine invertebrates representing four phyla and contrasting habitats, such as shallow rocky shores, typically growing in
Baltic waters, and deeper (about 30 m) muddy sediments in Atlantic waters, as well as behavioural strategies, including sessile (e.g. blue mussels) and mobile (e.g. hermit crab, sea starts and sea snails, Table 1). We hypothesize that responses could be driven by phyla and/or the habitat or niche the species occupy (Table 1). In particular, we expect all species to be able to cope with high $CO_2$, as they experience broad fluctuations of $CO_2$ in their habitat (Table 1), but should be vulnerable



to hypoxia, as they experience high $O_2$ levels in their habitat, except for those with an infaunal growth habitat, which are expected to be resistant to low $O_2$ and high $CO_2$, as in their habitat (Table 1). We also expect calcifiers to be particularly vulnerable to high $CO_2$ as additional energy to support calcification is required to cope with the reduced saturation state of carbonate minerals associated with high $CO_2$ (Hendriks et al., 2015). We used a two-way full factorial design enabling us to resolve additive and interactive effects.

We examined responses through survival and respiration rates, as down-regulation of metabolic rates has been proposed as a short-term evolutionary strategy to balance energy supply and demand when physiological processes are impaired by environmental stresses (Calosi et al., 2013). High $pCO_2$ and low $O_2$ imposes a significant strain on aerobic animals as the available energy acquired from oxic respiration can be reduced in the presence of increased $pCO_2$ (Brewer and Peltzer, 2009). This energy could otherwise be directed towards growth, reproduction, and other biologically and ecologically important activities. Reduced respiration is known to occur during hypoxia and both increases and decreases have been observed when pH is reduced (e.g. Whiteley, 2011; Wei and Gao, 2012). However, responses are often highly species-specific (Fabry, 2008; Malakoff, 2012; Calosi et al., 2013). Reduced metabolism is a beneficial response for organisms in the short-term, however could become problematic over extended periods (Melzner et al., 2009; Rosa et al., 2013), as they may be unable to produce the energy required to support key processes.

## 2    Methods and Materials

### 2.1    Site and Location

The experiments were conducted during August 2013 at the Sven Lovén Centre for Marine Sciences in Kristineberg, of the University of Gothenburg, Sweden (58° 14' 58" N and 11° 26' 44" E). The centre provided access to a diversity of marine life as it is located at the mouth of the Gullmar Fjord. This fjord is home to a mix of habitats with varying complexity and a salinity gradient of three distinct water masses: 1) the surface layer from the Kattegat Sea (salinity 24 – 27); 2) the more saline mid-waters (32 - 33) from the Skagerrak; and 3) the salty (34.4) North Sea water mass in the deeper sections of the fjord (Polovodova et al., 2011).

The fjord is home to a range of habitats such as sand and mud flats, seagrass beds, exposed and protected shorelines, and rocky bottoms, which together with the diversity of water masses results in high biodiversity (University of Gothenburg, 2011). It has a history of both natural and anthropogenically enhanced hypoxia and becomes vulnerable to these events when enrichment is high and flushing over the sill is slow, or does not occur at all (Josefson and Widbom, 1988; Arneborg, 2004). This area also has a history of North Sea upwelling, where salty water reaches the surface towards the end of summer and early autumn (see Fig. 2 and 3 in Lindahl et al., 2007).

### 2.2    Species, collection and maintenance

Specimens from 11 invertebrate species (Table 1) were collected from either surface or deep water within the Gullmar Fjord. *Ciona intestinalis* and *Littorina littorea* were collected by hand from mooring ropes and rocky shores, respectively, in the Grunsund boat harbour. *Asterias rubens* was also collected by hand from the rocky shore at the research station. All other specimens were retrieved with an Agassiz trawl aboard the research vessel *Oscar von Sydow* at up to 30 m depth over both rocky bottom and muddy sediment. *Amphuira filiformis* were collected with a 0.5 m sediment grab at 20 m depth. Only the top 10 cm of sediment from each grab was retained, as this was the oxygenated layer where organisms could be found. All organisms were maintained in flow-through tanks with (deep) North Sea water for at least two days before being placed into experimental aquaria. Water conditions followed the natural fluctuations occurring in the fjord (average pH



~ 8.0, salinity = 32.1 ± 0.02 ranging from 31.5 to 32.7, and temperature = 16°C ± 0.06 ranging from 14.1 to 17.3°C, data from http://www.weather.loven.gu.se/en/data).

Based on earlier experience in holding these species for experimental purposes, *Pagrus bernhardus* was fed, by allowing them to feed *ad libidum* on blue mussel meat, while being held in the tank prior to the experiment. *C. intestinalis*

and *L. littorea* were placed in plastic mesh cages (~ 0.5 cm$^2$) so that they were not lost through the outflow or escaped the aquarium. All gastropods, bivalves, and hermit crabs were cleaned with a toothbrush prior to use in order to remove any algae that could alter $O_2$ concentrations during measurements.

Invertebrates were exposed to one of the four treatments for a maximum of six days. Mortality events were rare across species (7 individuals died out of 168 used in the experiments) and insufficient to allow robust calculations of

mortality rates. Of these seven, three specimens (one each of *P. bernhardus*, *P. miliaris,* and *A. rubens*) died at the same time in the same aquarium indicating that there was likely an anomaly in the tank, although we could not determine its nature. The other four specimens that experienced mortality were *A. rubens* under $L_{O2}L_{CO2}$, and *P. bernhardus, M. edulis*, and *A. filiformis* under $L_{O2}H_{CO2}$. Survivorship in the control was 100%, 97.6% in $L_{O2}L_{CO2}$, and 92.9% in the $H_{O2}H_{CO2}$ and $L_{O2}H_{CO2}$ treatment.

### 2.3 Treatment protocol

Four treatments (3 replica aquaria each) with two levels of dissolved oxygen (DO) and $p$CO$_2$ concentration were used: a) $H_{O2}L_{CO2}$ – ambient $CO_2$ (400 µatm) and high $O_2$ (100% saturation or 9 - 10 mg L$^{-1}$); b) $L_{O2}L_{CO2}$ – ambient $CO_2$ and low $O_2$ (20 - 35% saturation or 2 - 3.5 mg L$^{-1}$); c) $H_{O2}H_{CO2}$ – high $CO_2$ (~1300 µatm) and high $O_2$; and d) $L_{O2}H_{CO2}$ - high $CO_2$

and low $O_2$.

The high $O_2$ aquaria were bubbled with ambient air, whereas the low $O_2$ aquaria were bubbled with a mixture of air and $N_2$ using an Aalborg GFC17 Mass Flow Controller (MFC) and a jar filled with glass marbles (allowing even mixing of gases) to create a mixture with reduced oxygen content. This was then bubbled through the six low $O_2$ treatments maintaining the DO between 2.0 - 3.5 mg L$^{-1}$, which was chosen after Vaquer-Sunyer and Duarte (2008)'s meta-analysis to

be a bit higher than the traditional definition of hypoxia by Diaz and Rosenberg (1995, 2008). The DO content of each aquaria was measured daily with PresSens oxygen micro-optodes (OXY 4 v2.11 Micro) that were calibrated in $O_2$ saturated deep-sea water (~10 mg DO L$^{-1}$ for 100% DO) and a 1 g ml$^{-1}$ sodium sulphite solution (0 mg DO L$^{-1}$ for 0% DO).

To increase the $p$CO$_2$, pure $CO_2$ was bubbled through high $CO_2$ aquaria. The low $O_2$ treatments also received $CO_2$ gas to maintain $CO_2$ at an ambient level due to the displacement of $CO_2$ in the presence of $N_2$. A reduction of 0.4 pH units

(equivalent to 1,300 µatm for high $CO_2$) from the ambient waters (at ~450 µatm in low $CO_2$) was chosen. These values correspond to the annual average atmospheric $p$CO$_2$ level for the high-end projected level for 2100 (IPCC, 2007) and for 2005, respectively. pH was controlled with Aqua Medic pH computers and 2.5W M-ventil valves. Each pH controller had a sensor attached to the aquarium, which opened the valve to release a burst of $CO_2$ when the pH was increasing beyond the set level (i.e. 7.6 or 8.0). The pH$_{NBS}$ (NBS scale) was measured daily of each aquaria (Consort datalogger D130 and

electrodes from Metrohm and Hanna Instruments) and every 3 days with a Metrohm 827 pH meter, calibrated at 15°C (with pH solutions of 3.99, 7.04, and 9.08).

Aquaria were continuously replenished by allowing deep water to flow through the tanks (filtered through a 20µm mesh) in a flow-through system with aquaria volume maintained at 17 L. Each replica aquaria held one individual from each species with the exception of *C. intestinalis* and *L. littorea*, which had two individuals per replica tank.

### 2.4 Carbonate chemistry

Gran titration method was used to measure total alkalinity (TA) every third day. Two 25 ml water samples were collected from each aquarium and filtered through a 45 µm filter. TA was measured at room temperature with a SI Analytics




TitroLine alpha plus machine and TitriSoft 2.6 software. Spectrophotometry was used to measure the $pH_{TOT}$ (total scale) at 25 °C of treatments with a Perkin Elmer Lambda 25 UV/VIS spectrometer and Perkin Elmer UV WinLab software to confirm the values of the daily $pH_{NBS}$ measurements (after Dickson, 2009). TA, $pH_{NBS}$, with temperature and salinity, were used to calculate the $pCO_2$, and aragonite and calcite saturation states ($\Omega_{arag}$ and $\Omega_{cal}$ respectively) in CO2SYS (Pierrot and

Wallace, 2006) with $K_1$ and $K_2$ constants from Mehrbach et al. (1973; refit by Dickson and Millero, 1987) and $KSO_4$ from Dickson (1990).

Respiration Index ($RI$) was calculated after Brewer and Peltzer (2009) as

$RI = \log10\ (pO_2/\ pCO_2)$                                             Eq. 1

where $RI \leq 0$ corresponds to the thermodynamic aerobic limit, a formal dead zone; at $RI = 0$ to 0.4 aerobic respiration does not occur; the range $RI = 0.4$ to 0.7 represents the practical limit for aerobic respiration, and the range $RI = 0.7$ to 1.0 delimits the aerobic stress zone (Brewer and Peltzer 2009).

### 2.5       Metabolic response

Respiration was measured on day three and six of exposure. Invertebrates were placed in pre-treated water for approximately five hours (depending on their size) in hermetically sealed containers. Oxygen was measured at the beginning and end of the incubation using the PresSens micro-optodes. A blank sample was measured to see if there was any natural 'drift'. Initial and final measurements were used to calculate the consumption rate standardized to dry weight (DW) as mg $O_2\ min^{-1}\ g\ DW^{-1}$. DW was measured after placing the individuals in the dry oven at 60˚C for at least 24 hours to remove any

moisture. All weight measurements were recorded with a Mettler Toledo AT261 Delta Range analytical balance (readability 0.01 mg).

The response ratio was calculated as the average metabolism in the experimental treatment ($X_E$), divided by the average metabolism in the control ($X_C$). The effect size for each treatment was the ln-transformed Respiration Rate (Kroeker

et al., 2010):

$Ln\ Effect\ Size = LnRR = \ln (X_E) – \ln (X_C),$                                Eq. 2

where $X_E$ and $X_C$ are the mean values of the response variable in the experimental and control treatments, respectively, where the control treatment was represented by the $H_{O2}L_{CO2}$ treatment. Bias-corrected bootstrapped 95% confidence interval was calculated after Hedges et al. (1999) and Gurevitch and Hedges (1999). The zero line indicates no effect, and

significance of mean effects is determined when the 95% confidence interval does not overlap zero.

### 2.6       Data analyses

One-way ANOVA's were conducted to test for differences in the respiration rate between treatments for each species. As there was no significant difference between time (i.e. difference between day 3 and 6) all data from days three

and six were pooled together. Where the respiration showed significant differences between treatments, a Student's t-test and post-hoc Tukey HSD test were conducted to resolve which treatments resulted in different respiration rates. A regression comparison was done to test the overall differences between the treatments. Moreover, a General Linear Model (GLM) was used to quantify species response to changes in $CO_2$, oxygen and their interaction. A significant, positive interaction term indicates synergistic effects between the stressors, while a significant, but negative interaction term implies antagonistic

effects, using the statistical software JMP (version 10.0; https://www.jmp.com) with the level for significance set at 0.05.





## 3        Results

### 3.1        Water conditions

The average measurements and calculated carbonate chemistry data for the experimental period are shown in
Table 2. On average (±SE) the targeted pH levels of 8.04±0.07 in low $CO_2$ treatments and 7.59±0.02 in the high $CO_2$
treatments were achieved, respectively, and significantly different from each other ($p < 0.0001$, ANOVA). The
corresponding atmospheric $CO_2$ levels were higher than our expected targets of 380 ppm ($H_{O2}L_{CO2}$ and $L_{O2}L_{CO2}$) and 1,000
ppm ($H_{O2}H_{CO2}$ and $L_{O2}H_{CO2}$). The desired average (±SE) oxygen content of 9.51±0.05 mg $L^{-1}$ for high oxygen treatments,
and 2.98±0.15 mg $L^{-1}$ for low oxygen treatments were also attained (Table 1; $p < 0.0001$, ANOVA). DO concentrations
remained relatively stable for the $H_{O2}L_{CO2}$ and $H_{O2}H_{CO2}$ treatments (SE = 0.06 for both) where 100% saturation was targeted.
DO concentrations in the $L_{O2}L_{CO2}$ and $L_{O2}H_{CO2}$ treatments were more variable ranging from 1.81 mg $L^{-1}$ up to 3.88 mg $L^{-1}$
over the course of the experiment. The pH was also most variable where manipulation was required in the $H_{O2}H_{CO2}$ (SD =
0.08 units) and $L_{O2}H_{CO2}$ (SD = 0.09 units) treatments, however there was also natural variation in the seawater as seen in the
$H_{O2}L_{CO2}$ and $L_{O2}L_{CO2}$ treatments (SD = 0.28). Overall pH and $O_2$ level were quite well maintained around the targeted
averages.

The *RI* averaged 1.60 ± 0.02 for the $H_{O2}L_{CO2}$, 1.15 ± 0.03 for the $L_{O2}L_{CO2}$, 1.14 ± 0.03 for the $H_{O2}H_{CO2}$ and 0.69 ±
0.04 for the $L_{O2}H_{CO2}$ treatment (Table 2). The *RI* values for the hypoxic and high $CO_2$ treatment were similar as the
differences in $pO_2$ and $pCO_2$ had a similar affect on *RI*. All treatments matched the target values and were held to an
acceptable level and variability within each treatment (Table 2).

### 3.2        Respiration

Although the overall response was not significant for any experimental treatment ($p = 0.357$; ANOVA), when
plotting the mean respiration rate of each species of the $H_{O2}L_{CO2}$ treatment versus the different experimental treatments (Fig.
1), results of regression analysis show that there is a significant difference between the 1:1 line in the $H_{O2}H_{CO2}$ treatment ($p <$
0.05; Regression comparison), whereas the other two treatments didn't differ significantly ($L_{O2}L_{CO2}$: $p = 0.701$; $L_{O2}H_{CO2}$: $p =$
0.070; regression comparison). When comparing results of the different habitats a significant difference between treatments
and habitats was observed ($p < 0.01$; two-way ANOVA), as the result of the mooring was different from the other three
habitats throughout treatments (Student's t).

The general trend for the $L_{O2}L_{CO2}$ and $L_{O2}H_{CO2}$ treatment was for organisms to reduce their metabolism, as
metabolic rates for most species fell below the 1:1 line (Fig. 1). The metabolic rate for *C. intestinalis* under ambient
conditions was over 2.5 times greater than that for any other species. Echinoderms generally displayed lower respiration
rates, with the exception of *A. filiformis* who had comparatively high metabolism (Fig. 2). The three species of molluscs had
similar metabolic rates, which differed amongst treatments.

When looking at the Ln Effect Size of each species separately, nine of the 11 species tested experienced reduced
respiration in response to the $L_{O2}L_{CO2}$ treatment, with six being significant different (Fig. 2; Table 3) compared to the
$H_{O2}L_{CO2}$ treatment. The species *A. filiformis* and *A. rubens* responded with increased respiration, although not significantly
(Fig. 2). As for the $H_{O2}H_{CO2}$, six species increased respiration, with a significant difference in *O. fragilis* and *M. edulis*. The
other five species responded with decreased metabolic rates (Fig. 2). The $L_{O2}H_{CO2}$ treatment also had quite variable results
with seven species experiencing lower respiration rates than the control, with significant differences in the species *A. rubens*,
*A. filiformis*, *L. littorea* and *T. granifera* (Fig. 2). The majority of species exposed to $L_{O2}L_{CO2}$, $H_{O2}H_{CO2}$, and $L_{O2}H_{CO2}$ did not
experience changes in respiration that differ significantly from those observed under $H_{O2}L_{CO2}$ conditions. This is confirmed





by the results of the GLM (Table 3), which showed that the responses to oxygen and $CO_2$ are highly species specific, as we observed synergetic effects in only four out of 11 species (*O. fragilis*, *O. nigra*, *A. rubens* and *T. granifera*; Table 3).

**4    Discussion**

The Baltic species tested were highly resistant to hypoxia and high $CO_2$, alone or in combination, as they experienced very high survival rate across treatments in the relatively short-duration experiment reported here. Whereas lethal responses to high $CO_2$ are seldom observed (Kroecker et al., 2013), the level of hypoxia imposed is sufficient to cause
mortality of half of the populations of most marine species (Vaquer-Sunyer et al., 2008), with elevated $CO_2$ expected to enhance respiratory stresses (Brewer and Peltzer, 2009). This suggests that the species tested have adapted to hypoxia and high $CO_2$, which are experienced regularly in the ecosystem (Table 1), as more vulnerable species would have been already removed from the community.

The resistance of all species tested to short-term (3 to 6 day exposure) hypoxia, high $CO_2$, and their combined
effects, reflected in negligible mortality rates and modest metabolic responses, suggest that the community in the Gullmar Fjord have already been sieved to contain species and lineages resistant to these stressors, to which they have been exposed, at least for short periods of time, for generations (Josefson and Widbom, 1988; Arneborg, 2004). Whereas physiological limits to low $O_2$ concentrations seem unavoidable (Brewer and Peltzer, 2009), the existence of thresholds for high $CO_2$ are less evident. Moreover, the fact that no or negligible mortality was experienced in our experiments at *RI*'s of 0.69, where
Brewer and Peltzer (2009) predict the organisms to be severely compromised, in the thermodynamic limit of aerobic respiration, supports the idea that organisms have acclimatized to reoccurring events of low $O_2$ (and low pH), which are well documented within the Gullmar Fjord (Rosenberg, 1985; Johannessen and Einar, 1996; Nordberg et al., 2000; Polovodova and Nordberg, 2013). While there is a relatively long history of monitoring in the Gullmar Fjord, one of the longest-studied ecosystems in the world (seawater temperature records exist since the 1700's), pH data collection has been erratic and often
only recorded at the surface (SMHI, 2011). However available data shows pH has fluctuated between 7.6 and 8.7 over the last six decades (Dorey et al., 2013).

Our experimental treatments explored a more limited range of $O_2$ and $CO_2$ than present across Gullmar Fjord. This area has a history of hypoxia and even complete anoxia within the last four decades (Nordberg et al., 2000; Polovodova et al., 2011), so that the community has already been sieved of species vulnerable to low $O_2$ concentrations. Indeed, our
$H_{O2}L_{CO2}$ values, involving saturating $O_2$ concentrations, are unlikely to be experienced at the fjords depths (Fig. 3). It is, therefore, possible that the low $O_2$ conditions better represented the environment in which the organisms were growing prior to the experiments. The experimental $CO_2$ values tested need also be compared with ambient levels. Dorey et al. (2013) found that pH in the Gullmar Fjord has varied between 8.7 and 7.6 over the last 66 years. Therefore the minimum pH level conducive to a rise in $pCO_2$ to 1,000 ppm would be closer to 7.2. Dorey et al. (2013) conducted lab experiments with pH
values as low as 6.5 for urchin larvae, which are generally more sensitive to pH change than adults (Dupont and Thorndyke, 2009). Hu et al. (2014) tested *A. filiformis* living in sediment burrows with a pH of 7.0, which also happened to have $O_2$ levels below 2.0 mg L$^{-1}$. Hence, the community tested here already has $O_2$ and pH values comparable to those used as treatments here, particularly for infauna, such as *A. filiformis* and *B. lyrifera* which appear to be exposed to low $O_2$ and pH conditions on a regular basis.

Sublethal responses, in terms of metabolic depression or enhancement, were observed in response to hypoxia and high $CO_2$, alone or in combination. We expected that $L_{O2}H_{CO2}$ would be the most significant respiratory stress for organisms, as it would affect all species except those with an infaunal growth habit, and thus would result in a reduced metabolism. However, only one species (*A. filiformis*) with an infaunal growth habit (Table 1) experienced a significantly reduced



metabolism due the coupled impacts of $H_{CO2}$ and $L_{O2}$. Two of the three species with an infernal growth habit showed no metabolic response to hypoxia, whereas all except two of the species growing in other habitats, generally experiencing high oxygen levels, experienced a metabolic depression under hypoxia. Whereas there were no consistent patterns in the responses across phyla, they showed consistency among habitats, reflecting the conditions the species were adapted to in

their natural habitat.

There is growing interest in understanding the response of marine organisms to multiple stressors such as rising temperature, OA, increased UVB radiation, and reduced $O_2$ (e.g. Pörtner et al., 2005; Fredersdorf et al., 2009; Vaquer-Sunyer and Duarte, 2010; 2011, Duarte, 2014). For example, Mayol et al. (2012) examined the co-occurrence of low $O_2$ with high $CO_2$ in the Pacific Ocean off the Chilean coast, identifying layers where both stressors co-occur. Yet, most

experimental evidence of the response of marine invertebrates to stressors focus on individual effects, where their combined effects may differ from those derived (or calculated) from combinations of individual effects (cf. in Kroeker et al., 2013). Indeed, multiplicative, rather than additive, effects of the impacts of the individual stresses are expected (Pörtner et al., 2005; Clapham and Payne, 2011; Ateweberhan et al., 2013). However, our results demonstrate that there is a broad range of possible impacts within species from the Gullmar Fjord ecosystem including species that show an amplification of the

responses beyond that expected under an additive model and those that show a buffering or compensation of responses when multiple stressors co-occur. The *A. rubens* exhibited a synergistic response to hypoxia and high $CO_2$ as it showed a significant metabolic depression under both stressors, but no significant response to either one alone. The echinoderm *O. fragilis* experienced enhanced metabolic rates when exposed to high $CO_2$, consistent with the sensitivity to high $CO_2$ reported for their larvae, which experienced 100 % mortality when pH was reduced by just 0.2 units (Dupont and

Thorndykee, 2008). In contrast, *M. edulis* experienced depressed metabolism when exposed to hypoxia. As a result, these effects operated into an antagonistic mode, resulting in no significant change in metabolic rates when the organisms were exposed to both hypoxia and high $CO_2$. However, there was no general trend for responses to be either additive or synergistic across species. Indeed, our result suggests that responses are mostly dependent on the fluctuations in the stressors in their habitats, so that the prior selective and adaptive history of the species plays an important role in determining their

vulnerability to different stressors.

Whereas a theoretical framework to predict the response of marine organisms to multiple stressors is generally lacking, Brewer and Pelzer (2009) derived a theoretical expectation of the expected responses in the particular case of combined hypoxia and high $CO_2$, the organisms tested show a *RI* decrease with intensity of alterations in our treatments as expected. Although we reached the 0.7 threshold value (0.69 under $L_{O2}H_{CO2}$), which represents the thermodynamic limit for

aerobic respiration, the organisms are expected to be severely compromised. Yet, we observed little or no mortality and the organisms exposed to $L_{O2}H_{CO2}$ should have experienced aerobic stress, yet, our results showed that they were more likely to reduce respiration under hypoxia. Hence, the *RI* does not appear to hold predictive power as to the response of marine invertebrates to the interactions between $O_2$ and $CO_2$. All but one of the tested species were calcifiers, and were expected to be impacted by high $CO_2$. Indeed, the high $CO_2$ treatment reached an undersaturated concentration of aragonite ($\Omega_{arag} < 1$),

where calcifiers are expected to be stressed (Doney et al., 2009). Molluscs rely chiefly on aragonite to construct their shells (Porter, 2007), while echinoderms and crustaceans use calcite (Raup, 1959; Raabe et al., 2005). Yet, the impacts of high $CO_2$ were not greater in molluscs than for echinoderms and crustaceans in our experiments. Hence, neither the Respiration Index nor a threshold at $\Omega_{arag} < 1$ for calcifiers appeared to hold predictive power on the effects of hypoxia and/or $CO_2$ on the species tested here, which seemed best predicted from consideration of the ranges of $O_2$ and $CO_2$ they experience in their

habitat.

Responses to low $O_2$ and high $CO_2$ were variable amongst phyla and species in the community tested here, ranging from antagonistic to synergistic responses. The very limited impacts of low $O_2$ and high $CO_2$ of the invertebrates from this ecosystem, which showed little or no mortality in the presence of both stressors, reflects the range of conditions in





the habitats these organisms occupy. This ecosystem has been reported to experience recurrent hypoxic events characterised by low pH values and high $CO_2$ (Nordberg et al., 2000; Dorey et al., 2013). Hence, the organisms tested were resistant to both stressors within the levels used in this experiment, which, while ranging within values reported to negatively impact on marine invertebrates for both $O_2$ (Vaquer-Sunyer and Duarte, 2008) and $CO_2$ (Kroeker et al., 2013), were within the range

present in their ecosystem. Eutrophication-driven hypoxia, such as experienced in Baltic fjords, derives from excess metabolic $O_2$ consumption and is, therefore, often associated with elevated $CO_2$ (e.g. Duarte et al., 2013, Melzner et al., 2013, Wallace et al., 2014). Hence, low $O_2$ and high $CO_2$ often co-occur in areas affected by hypoxic events, such as Gullmar Fjord. Haselmair et al. (2010) observed that pH declined by up to 0.7 units during an induced anoxic event in the Adriatic Sea and Melzner et al. (2013) predict that $p$CO$_2$ can reach up to 3,200 ppm during anoxic events in brackish waters

(salinity of 20), with those values decreasing as salinity increases. Hence, adaptive responses of organisms in the Gullmar Fjord should be coupled for low $O_2$ and high $CO_2$, thereby accounting for the limited effects to the experimentally imposed stressors used here.

**5      Conclusions**

Responses to low $O_2$ and high $CO_2$ were variable amongst phyla and species in the community tested here, ranging from buffered to amplified metabolic responses. The very limited impacts of low $O_2$ and high $CO_2$ of the invertebrates from this ecosystem, which showed little or no mortality in the presence of both stressors, reflects the past history of this ecosystem, which has been reported to experience recurrent hypoxic events characterised by low pH values

and high $CO_2$ (Nordberg et al., 2000; Dorey et al., 2013). Hence, the organisms trialled were resistant to both stresses within the levels used in this experiment, which were within values reported to negatively impact on marine invertebrates for both $O_2$ (Vaquer- Sunyer and Duarte, 2008) and $CO_2$ (Kroeker et al., 2013). Hypoxia impacted the greatest number of organisms and represents, therefore, the most concerning stress in the region. Management plans addressing hypoxia should also avoid the generalized assumption that synergistic stressors will result in multiplicative effects and focus research into

understanding the mechanisms calcifiers and other invertebrates employ to cope with these changes. Our results also highlight the idiosyncratic nature of responses, which were strongly species-specific, suggesting that extrapolations from experiments conducted on a few species to the phylum level may be strongly misleading. This adds complexity to the challenge of predicting how global stressors will affect marine ecosystems in the future.

**Author contribution**

design of the experiment: AF, AS, CMD

experimental part: AF, AS, SD

analysis: AF, AS, SD, CMD

writing: AF, AS, SD, CMD

**Competing interests**

The authors declare that they have no conflict of interest.



**Acknowledgements**

This research was funded by projects ASSEMBLE (grant agreement no. 227799; under the EU Research Infrastructure Action FP7) and the Estres-X project funded by the Spanish Ministry of Economy and Competitiveness (CTM2012-32603). A. Fontanini was funded by the School of Plant Biology at the University of Western Australia (grant 10300374) and A. Steckbauer was funded by a fellowship from the Government of the Balearic Islands (Department on Education, Culture and Universities) and the EU (European Social Fund) as well as King Abdullah University of Science and Technology. SD is funded by the Linnaeus Centre for Marine evolutionary Biology at the University of Gothenburg and supported by a linnaeus grant from the Swedish research Councils VR and Formas. We thank K. Chan, P. Engström and J. Dombrowski for assistance.



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





**Table 1.** Species used in the experiment along with the habitat where they were collected and the characteristic pH and $O_2$ levels in these habitats. pH and $O_2$ values at the habitats from Dorey et al. (2013), Hu et al. (2014), and Grans et al. (2014).

| Species | Sampling site | pH | Oxygen (mg L$^{-1}$) |
|---|---|:---:|:---:|
| *Ciona intestinalis* | Mooring rope, surface | Highly variable (8.7-7.6) | High (8) |
| *Pagurus bernhardus* | Gravels, 30m | Variable (8.1-7.7) | High (8) |
| *Littorina littorea* | Rocky shore, surface | Highly variable (8.7-7.6) | High (8) |
| *Tarebia granifera* | Soft sediment, 30m, infaunal | Low (7.6) | Low (1.6) |
| *Mytilus edulis* | Rocky shore, surface | Highly variable (8.7-7.6) | High (8) |
| *Ophiocomina nigra* | Gravels, 30m | Variable (8.1-7.7) | High (8) |
| *Ophiothrix fragilis* | Gravels, 30m | Variable (8.1-7.7) | High (8) |
| *Amphiura filiformis* | Soft sediment, 30m, infaunal | Low (7.6) | Low (1.6) |
| *Asterias rubens* | Rocky shore, surface | Highly variable (8.7-7.6) | High (8) |
| *Psammechinus miliaris* | Gravels, 30m | Variable (8.1-7.7) | High (8) |
| *Brissopsis lyrifera* | Soft sediment, 30m, infaunal | Low (7.6) | Low (1.6) |





**Table 2.** Realised carbonate chemistry and oxygen concentrations for the four treatments ($H_{O2}L_{CO2}$, $L_{O2}L_{CO2}$, $H_{O2}H_{CO2}$, and $L_{O2}H_{CO2}$). Values are averages ± SE of measurements and calculations (using CO2SYS). Respiration Index (RI) as defined by Brewer and Peltzer (2009) (see section 2.4 for details).

| | $H_{O2}L_{CO2}$ | | $L_{O2}H_{CO2}$ | | $H_{O2}H_{CO2}$ | | $L_{O2}H_{CO2}$ | |
|---|---|---|---|---|---|---|---|---|
| | Mean | SE | Mean | SE | Mean | SE | Mean | SE |
| Temperature (°C) | 15.4 | 0.2 | 15.4 | 0.2 | 15.4 | 0.2 | 15.4 | 0.2 |
| Oxygen (mg L$^{-1}$) | 9.43 | 0.06 | 2.91 | 0.20 | 9.60 | 0.06 | 3.05 | 0.24 |
| $pH_{NBS}$ | 8.02 | 0.01 | 8.06 | 0.02 | 7.58 | 0.03 | 7.61 | 0.04 |
| Salinity | 32.16 | 0.12 | 32.16 | 0.12 | 32.16 | 0.12 | 32.16 | 0.12 |
| Total Alkalinity (µmol kg$^{-1}$) | 2245.9 | 11.2 | 2249.5 | 9.1 | 2249.5 | 7.2 | 2258.8 | 11.5 |
| $pCO_2$ (µatm) | 591.9 | 22.4 | 536.7 | 26.6 | 1783.5 | 131.7 | 1687.7 | 146.7 |
| $HCO_3^-$ (µmol kg$^{-1}$) | 1975.4 | 15.9 | 1951.8 | 16.7 | 2143.6 | 9.0 | 2144.8 | 16.6 |
| $CO_3^{2-}$ (µmol kg$^{-1}$) | 111.3 | 3.0 | 120.4 | 4.1 | 44.5 | 2.9 | 47.6 | 3.5 |
| Ω Aragonite | 1.73 | 0.05 | 1.87 | 0.06 | 0.69 | 0.04 | 0.74 | 0.05 |
| Ω Calcite | 2.71 | 0.07 | 2.93 | 0.10 | 1.08 | 0.07 | 1.16 | 0.08 |
| *RI* | 1.60 | 0.02 | 1.15 | 0.03 | 1.14 | 0.03 | 0.69 | 0.04 |



**Table 3.** Respiration Rate (± SE) and results of the General Linear Model (GLM) off all tested species (pooled data where we had 3 and 6 days). Levels not connected by the same letter are significantly different (after Student's T and Tukey HSD tests). Numbers written in red color highlight significant differences.

| Species | Taxa | day | Prob. > F | | average respiration rate (± SE) $H_{O2}L_{CO2}$ | $L_{O2}L_{CO2}$ | $H_{O2}H_{CO2}$ | $L_{O2}H_{CO2}$ | General Linear Model (GLM) $CO_2*O_2$ |
|---|---|---|---|---|---|---|---|---|---|
| *Pagurus bernhardus* n = 12 | Crustacean | 3/6 | 0.0417 | Average (± SE) Students' T Tukey HSD | 0.067 0.010 A AB | 0.043 0.003 B B | 0.054 0.010 AB AB | 0.073 0.005 A A | -0.0434 |
| *Ciona intestinalis* n = 24 | Tunicata | 3/5 | 0.1578 | Average (± SE) Students' T Tukey HSD | 0.265 0.025 A A | 0.236 0.050 A A | 0.366 0.052 A A | 0.236 0.049 A A | 0.1001 |
| *Brissopsis lyrifera* n = 12 | Echinoidea | 3 | 0.0715 | Average (± SE) Students' T Tukey HSD | 0.0058 0.0005 A A | 0.0027 0.0012 B A | 0.0067 0.0008 A A | 0.0046 0.0011 AB A | -0.0009 |
| *Psammechinus miliaris* n = 12 | Echinoidea | 3/6 | 0.1202 | Average (± SE) Students' T Tukey HSD | 0.024 0.002 A A | 0.016 0.001 B A | 0.022 0.004 AB A | 0.023 0.003 AB A | -0.0090 |
| *Amphiura filiformis* n = 12 | Echinoidea | 3 | 0.1678 | Average (± SE) Students' T Tukey HSD | 0.115 0.035 AB A | 0.172 0.045 A A | 0.081 0.019 AB A | 0.046 0.023 B A | 0.0904 |
| *Ophiothrix fragilis* n = 12 | Echinoidea | 3 | 0.0023 | Average (± SE) Students' T Tukey HSD | 0.0098 0.0014 B BC | 0.0023 0.0004 C C | 0.0204 0.0048 A A | 0.0115 0.0011 B AB | 0.0015 |
| *Ophiocomina nigra* n = 24 | Echinoidea | 3 2/4/6 | 0.2054 | Average (± SE) Students' T Tukey HSD | 0.013 0.001 AB A | 0.012 0.001 B A | 0.015 0.001 A A | 0.014 0.001 AB A | 0.0006 |
| *Mytilus edulis* n = 12 | Bivalve | 3/6 | 0.0063 | Average (± SE) Students' T Tukey HSD | 0.027 0.002 A AB | 0.011 0.003 B B | 0.043 0.008 A A | 0.030 0.008 A AB | -0.0045 |
| *Asterias rubens* n = 12 | Gastropoda | 3 | 0.3302 | Average (± SE) Students' T Tukey HSD | 0.042 0.002 A A | 0.042 0.011 A A | 0.032 0.007 A A | 0.019 0.013 A A | 0.0133 |
| *Littorina littorea* n = 24 | Gastropoda | 3/6 | < 0.0001 | Average (± SE) Students' T Tukey HSD | 0.083 0.006 A A | 0.040 0.006 B C | 0.067 0.007 A AB | 0.050 0.004 B BC | -0.0254 |
| *Tarebia granifera* n = 12 | Gastropoda | 3/6 | 0.0073 | Average (± SE) Students' T Tukey HSD | 0.007 0.002 AB AB | 0.005 0.001 B B | 0.011 0.002 A A | 0.004 0.000 B B | 0.0044 |



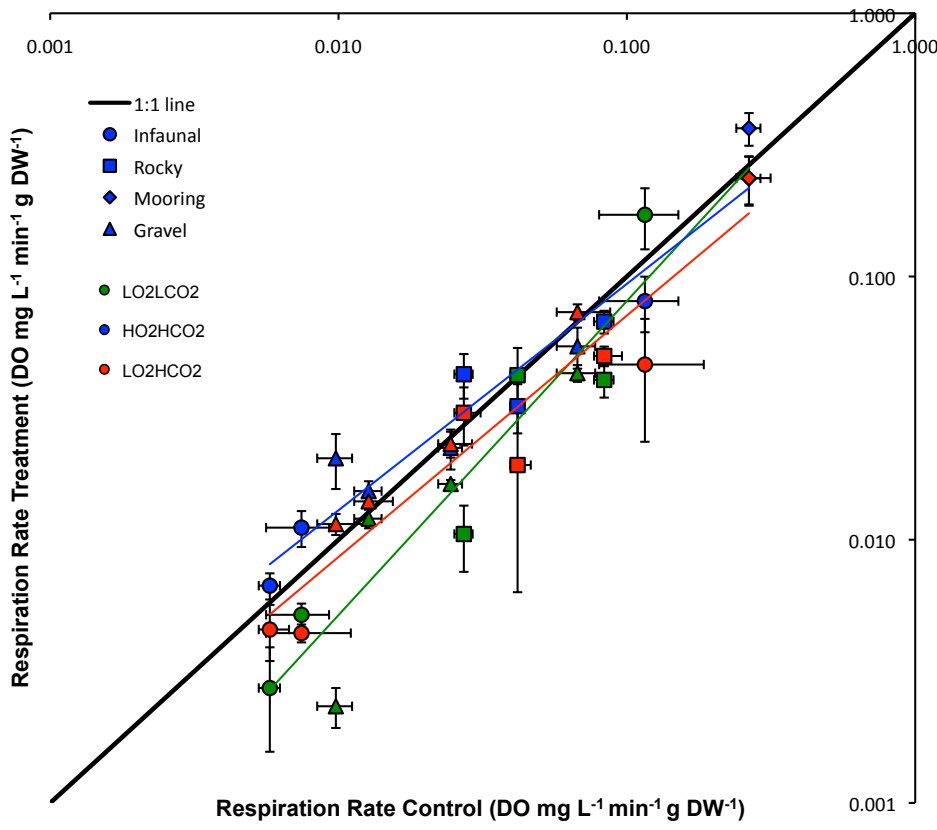

**Figure 1:** Respiration rate (Average ± SE) control vs. treatments of all tested species: green – $L_{O2}L_{CO2}$ (y = 0.9571x − 0.0046, $R^2$ = 0.90),
blue – $H_{O2}H_{CO2}$ (y = 1.2905x − 0.0122, $R^2$ = 0.93) and red – $L_{O2}H_{CO2}$ (y = 0.8301x − 0.0032, $R^2$ = 0.92). The 1:1 line represents where
10  treatment metabolism is equal to ambient metabolism.



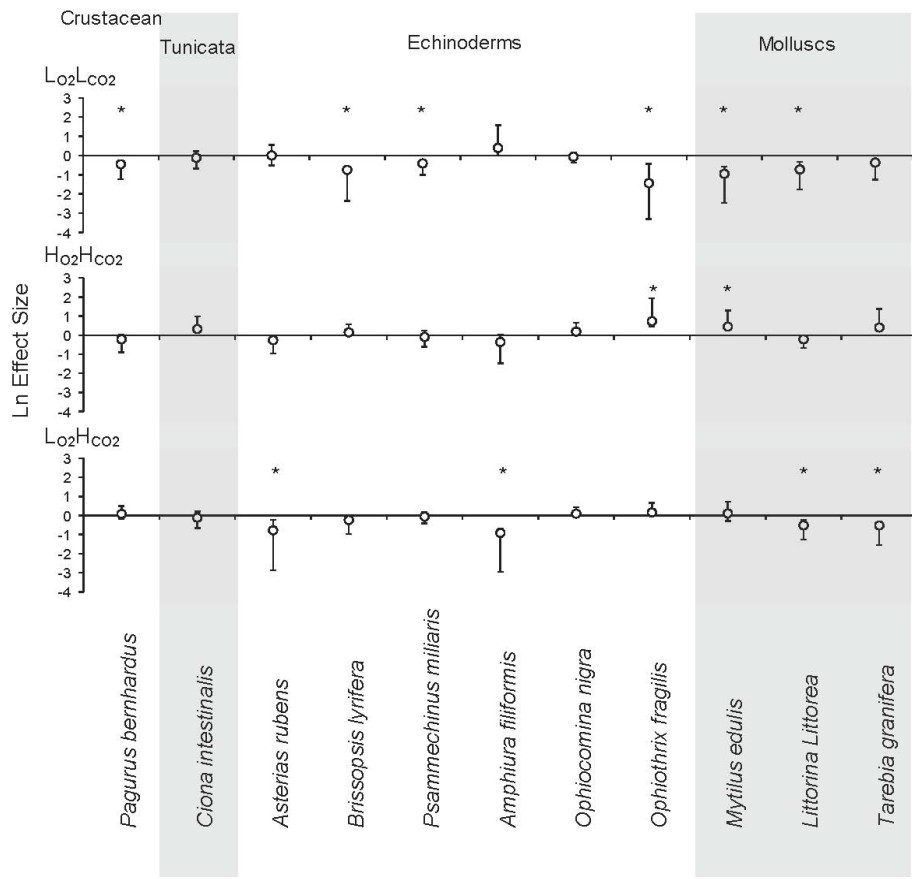

**Figure 2:** The Ln Effect Size of the response ratios for invertebrate species and phyla in response to three treatments: low $O_2$ ($L_{O2}L_{CO2}$), low pH ($H_{O2}H_{CO2}$), and coupled low $O_2$ and low pH ($L_{O2}H_{CO2}$) compared to control levels ($H_{O2}L_{CO2}$). LnRR = ln(treatment)-ln(control) ±

10   Bias-corrected bootstrapped 95% confidence interval (after Kroeker et al., 2010; Hedges et al., 1999; Gurevitch and Hedges, 1999). The zero line indicates no effect, and significance of mean effects is determined when the 95% confidence interval does not overlap zero (significant results marked with '*'). Grey background was added to summarize the species by phyla.



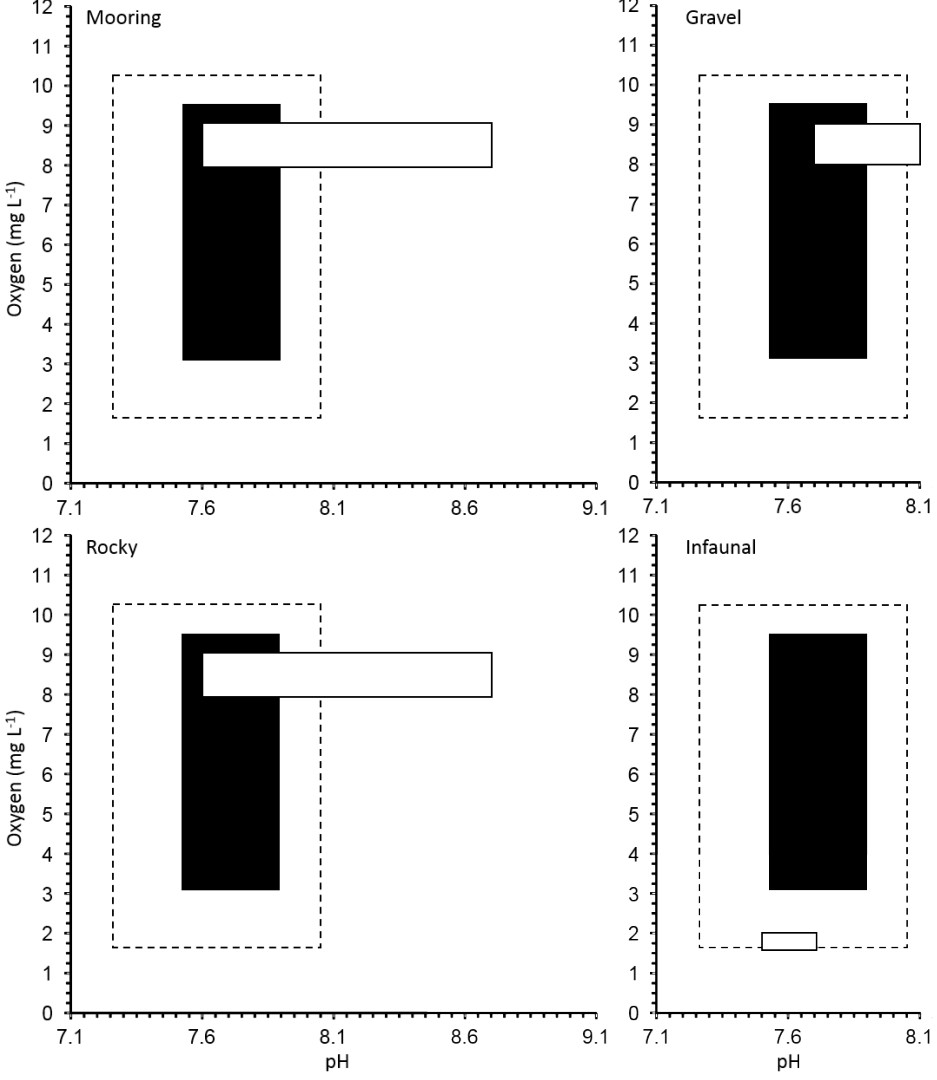

**Figure 3:** Realised oxygen concentration and pH conditions for a manipulation experiment for 11 invertebrate species from four different natural habitats in the Gullmar Fjord; gravel, infaunal, mooring & rocky. Average and extreme (maximum & minimum) $O_2$ and pH conditions during experimental exposure are show in black and with dotted line, respectively. The natural $O_2$ and pH conditions expected for each habitat are shown in the white boxes.

