# Peer review of "Variable metabolic responses of Skagerrak invertebrates to low O2 and high CO2 scenarios"

_Biogeosciences, 2017_

## Referee Comment (RC1) · Anonymous Referee #1 · 14 Nov 2017

Fontanini and co-wokers investigated the respiratory responses of a range of invertebrates to short-term changes in oxygen and CO2. The paper gives some interesting insights which revealed the highly variable responses of invertebrates from a single area towards future scenarios. This reveals that e.g. general ecosystem models based on few species are not yet reliable. Therefore this study highlights the need for more experimental work. Currently the manuscripts lack some important experimental details which make it complicated to judge the quality of the measurements. For instance what was the rational behind measurements on day 3 and 6 and were any differences observed? Considering the strong physiological differences between the investigate phyla e.g. a crustacean and a tunicate or echinoderm it is not intuitive why the animals were grouped according to the habitat and not according to the lifestyle in figure 1. This

could mask certain significances. I have a couple comments and the MS would benefit form some careful corrections thus I recommend major revision.

Specific comments Page 1 Line 25 that the responses of respiration of the respiration – please change

P 2 line 6 please change to acidic water

P2 Line 7 what do you mean by 'Control'? more precisely speaking: the biochemical processes which change seawater pH?

P2 Line 18 Internal is not a precise term, intra- and extracellular pH regulation are two completely different processes. I assume you refer to extracellular pH as intracellular pH is commonly well regulated? Please specify.

P2 Line 20 Hemoglobin is not common in invertebrates which this study is focused on.

P3 Line 31 what is meant by 'a history of North Sea upwelling'? commonly observed?

P3 line 3 and following? Where the salinity similar at surface and bottom of the Fjords as all animals were exposed to the same high salinity during the acclimation phase?

P4 Line 3 and following Based on this paragraph, P. bernhardus was the only species which was fed during the experiment? Is there any specific reason for this decision?

P4 Line 8 and following Several species and sometimes specimens were kept in the same aquaria for logistic purposes? However can you exclude that a number of co-variable influenced respiration rates similar to the observed abnormal mortality in one tank?

P4 Line 16 and following The animals were kept in closed systems without any waster exchange. Did you check the water quality in order to monitor potential accumulation of waste products due to metabolism and mortality?

P 5 Line 15 and following As respiration was only response variable measured in this

study, more detailed information needs to be provided such as: Volume of the containers, did you control for a linear decline of oxygen concentrations? In particular, as this is the focus of the study, how much did oxygen decline during the incubation? Strong declines would severely affect the study concept.

P 4 line 34 Why did you use two different pH meters and what differences did you observe?

P 6 line 7 Here you state a target of 1000 ppm whereas it is 1300 $\mu$atm in the M&M, even if target and measurement were not identical, the target should be uniform.

P6 Line 37 A non significant response may only called a 'trend towards'

P 7 line 7 the experiment did not last long enough to draw any reliable conclusion on survival rates

P7 line 14 To support this hypothesis you need to add a reference which documents higher mortality for populations from habitats with less abiotic stress

P7 line 17 and following Please consider that the RI hypothesis and in particular the definition of an exact threshold is still under debate: https://www.biogeosciences.net/10/2815/2013/bg-10-2815-2013.pdf

P8 Line 37 Even though calcified structures and the calcification process might be affected by undersaturation it is not clear why this should be detectable in the rates of aerobic metabolism

P9 line 6 Hypoxia is necessarily always coupled to elevated CO2

Table 3 please give the unit for respiration rates

Some references are either missing in the text or in the reference list e.g. Grans or Gräns? et al. is not in the list

---

## Referee Comment (RC2) · Anonymous Referee #2 · 13 Dec 2017

General comments The topic of the present work is interesting and the authors try to make a correlation between the obtained data and the environmental characteristics of habitats. However, my concern is the period of exposure to particular stressors, which may determine and the range of tolerance of marine invertebrates to environmental changes. For example it is reported (page 8, line 43) that invertebrates from this ecosystem which showed little or no mortality in the presence of both stressors, reflects the range of conditions in the habitats these organisms occupy. However, mortality is depended on several factors including reproduction period, body size, etc. Moreover it is depended on the period of species exposure to stressors. Also, a key point for the species to withstand stressful conditions for long term is their ability to keep a stable energy turnover since. Such metabolic responses and patterns determine and their

thermal limits. Even more some species live at the edges of their range of thermal tolerance. Thus long-term experiments might help further not only in estimating species' ability to withstand stressful conditions but to make a better correlation with the future climate projects. I consider that the authors should take into their consideration the above points and to reconsider the interpretation of some obtained data. I agree with author's statement about the complexity of stressors and the challenge of predicting how global stressors will affect marine ecosystems in the future.

Specific comments Page 2, line 8. This metabolic control. . .could be changed to The involvement of metabolic processes in the regulation of the pH in coastal water is... . .. Page 2, line 15. . . ...although the combined stress from depleted $O_2$ and high $CO_2$ is likely to provide a significant challenge to coastal invertebrates and less mobile species. . . could be changed to although the synergistic effect of $O_2$ depletion and $CO_2$ accumulation is likely to provide a significant challenge to coastal invertebrates and mostly to sessile species. Page 2, lines 24-26. There are many invertebrates tolerant to hypoxia (e.g. mussels). Thus the authors should be focused on these species which rather are less tolerant (e.g. benthic invertebrates). Page 2, line 37. I would prefer synergistic instead joint Page 2, line 38. ..future levels of what I consider that the two last paragraphs should be reorgasinized and rewritten in such a way so the firstly the authors to be reported at several hypotheses and secondly at their aims Methods and Materials 1. Merge the two first paragraphs 2. Make clear, when saying history, whether the reported environmental characteristics are long lasting. It is very important since species experiencing such environmental changes in their life cycle may have adapted to such environmental changes by developing the corresponding cellular and physiological mechanisms. 3. Report which of the examined rocky species are exposed or not to air because of tide. The latter characterizes sessile species tolerant to hypoxia. 4. Change Metabolic response to Metabolic rate or Oxygen consumption. Metabolic responses usually is referred when we examine the metabolic patterns (e.g. enzyme activities, metabolites etc) 5. Page 5, line 8. For the readers describe briefly the physiological meaning of term respiration index. 6. Page 5, lines 15-21. The experimental procedure for determining the oxygen consumption should be written in details. For example chamber volume, was it the same for all species examined? Also report the temperature, salinity and pH of water. It is very important to report the period (hours) of experimental procedure since under a particular level of PO2 metabolism sifts from aerobic to anaerobic and this point is species depended. 7. Page 5, line 23. Ratio of what? Results 1. Respiration. Report the consumption of oxygen rate for each examined species and give possible differences between each other. 2. Give more information the differences or not for the oxygen consumption for each species at each treatment 3. In general the results should be rewritten in such a way so to be more clear what is happening in each species at tested treatments and whether differences were recorded from species to species. Table 3. In the column day it is marked 3/6, 3/5 etc. In the legend it is reported pooled data where we had 3 and 6 days. Thus the number 4, 5 2 what do they mean.

Discussion Page 7, lin e 29-30. It is unclear what the authors report. Page 7, line 32-33. It is very important to report whether such changes in pH regard fluctuations or permanent changes. In the first case the organisms face waves of such changes and how long such waves last. Page 7, line 36-37. Community of what? Rewrite the sentence (line 37-39), since it is unclear what it is meaning. Page 9, line 16. Responses . . .which responses? Page 9, line 1-2. Do you know how long these events last? Is it an acute environmental change or long-term change? Page 9, line 10. It could be nice if the authors could support such adaptive responses, genetically determined, by reporting differences from individuals of the same species but from different populations habiting environments differing in the tested abiotic factors. The observed responses in the present work may regard phenotypic plasticity which may be observed and in individuals from populations living in other environments when treated similarly.

---

## Author Comment (AC1) · 25 Jan 2018

Fontanini and co-workers investigated the respiratory responses of a range of invertebrates to short-term changes in oxygen and CO2. The paper gives some interesting insights which revealed the highly variable responses of invertebrates from a single area towards future scenarios. This reveals that e.g. general ecosystem models based on few species are not yet reliable. Therefore, this study highlights the need for more experimental work. Currently the manuscripts lack some important experimental details which make it complicated to judge the quality of the measurements. For instance what was the rational behind measurements on day 3 and 6 and were any differences observed? Considering the strong physiological differences between the investigate phyla e.g. a crustacean and a tunicate or echinoderm it is not intuitive why the animals

were grouped according to the habitat and not according to the lifestyle in figure 1. This could mask certain significances. Reply: The decision about the duration of the experiment was based on a meta-analysis by Vaquer-Sunyer & Duarte (2008) which found the median lethal time (LT50) for over 400 studies to be just over 5 days. Therefore, we felt we should be able to detect a sub-lethal response such as respiratory changes within that time-frame. Moreover, as we wanted to measure the respiration rate, we had to make sure that the individuals were still alive. We have added a statement to the introduction to highlight this. The habitat background was the same for the species collected from the same habitat. Most of the tested species were echinoderms and gastropods. As for crustaceans, bivalves and tunicates, we just had 1 species each and thus didn't see the need of grouping per taxa.

I have a couple comments and the MS would benefit from some careful corrections thus I recommend major revision. Specific comments Page 1 Line 25 that the responses of respiration of the respiration – please change Reply: Changes made as suggested.

P 2 line 6 please change to acidic water Reply: We changed the phrase and it is now reading "leading to a decreased pH" as we try to avoid the term "acidic water".

P2 Line 7 what do you mean by 'Control'? more precisely speaking: the biochemical processes which change seawater pH? Reply: Yes, we were referring to the biochemical processes and relationships that may cause pH to fluctuate over various temporal and spatial scales, drawing particular attention to the role of metabolism. This sentence has been altered to better reflect this. The sentence now reads "The involvement of metabolic processes in the regulation of pH in coastal water is particularly evident when eutrophication stimulates algal CO2 for marine organisms reach further than the highly documented impacts on calcification rates (Doney et al., 2009).".

P2 Line 18 Internal is not a precise term, intra- and extracellular pH regulation are two completely different processes. I assume you refer to extracellular pH as intracellular pH is commonly well regulated? Please specify. Reply: Yes, we agree and changed

the phrase to "extracellular acid-base regulation".

P2 Line 20 Hemoglobin is not common in invertebrates which this study is focused on. Reply: We have removed the reference to haemoglobin as well as the definition of hypercapnia as we do not investigate the impact on blood fluids within this study.

P3 Line 31 what is meant by 'a history of North Sea upwelling'? commonly observed? Reply: Salinity at the surface of the fjord can change by 10psu in the summer months as salty water originating from the North Sea comes to the surface. We have decided to remove this sentence as the water we used during experiments had a stable salinity for all animals.

P3 line 3 and following? Where the salinity similar at surface and bottom of the Fjords as all animals were exposed to the same high salinity during the acclimation phase? Reply: The Sven Loven Centre has three water inflows from different depths in the Fjord. We chose to use North Sea (deep water) as animals from shallow environments are able to cope with the salinity due to the aforementioned upwelling events. As we have removed the reference to summer upwelling, we have also removed the reference to North Sea water from this sentence. This will have minimal impact as we have stated the salinity over the course of the experiment in subsequent sentences.

P4 Line 3 and following Based on this paragraph, P. bernhardus was the only species which was fed during the experiment? Is there any specific reason for this decision? Reply: Reply: This decision was based on advice from the lab technician in charge of looking after animals at the facility, which considerably experience in maintaining the organisms tested here. Given the short exposure periods, and the fact that most individuals are filter feeders, there was no need to add food to the aquaria. The crustacean P. bernardus on the other hand is a carnivore and was fed ad libidum before the experiment started. We added the sentence "No animals were fed during their experimental period" to clarify.

P4 Line 8 and following Several species and sometimes specimens were kept in the

same aquaria for logistic purposes? However, can you exclude that a number of co-variable influenced respiration rates similar to the observed abnormal mortality in one tank? Reply: Respiration rates of animals were measured individually in glass chambers, where no other organisms were present. All aquaria had the same 'mixture' of organisms at the same time, so we would expect to see the impact of co-variables across all treatments. Moreover, we made sure no predator and preys were in the tanks at the same time to exclude stress due to their presence.

P4 Line 16 and following The animals were kept in closed systems without any waster exchange. Did you check the water quality in order to monitor potential accumulation of waste products due to metabolism and mortality? Reply: Should that read Page 5 Line 16? As described on Page 5 Line 1, we replenished the water in the aquaria with a continuous flow of water. For the incubations to measure the respiration rate it was essential to keep the glass chambers/containers sealed as we measured oxygen at the beginning and end of the incubation. Any water-exchange would have influenced and changed those measurements.

P 5 Line 15 and following As respiration was only response variable measured in this study, more detailed information needs to be provided such as: Volume of the containers, did you control for a linear decline of oxygen concentrations? In particular, as this is the focus of the study, how much did oxygen decline during the incubation? Strong declines would severely affect the study concept. Reply: Oxygen consumption has been updated to $mg\ L^{-1}\ O_2\ min^{-1}\ L^{-1}\ g\ DW^{-1}$ as the different size of glass chambers was accounted for when calculating the respiration rates.

P 4 line 34 Why did you use two different pH meters and what differences did you observe? Reply: The Metrohm 827 pH meter was actually used to take daily point measurements and is shown in the information in the results section. The data logger was used as a reference for us to see what was happening overnight, but could only be placed in one tank and so has not contributed to any data reflected in this paper. We have removed the reference to the data logger.

P 6 line 7 Here you state a target of 1000 ppm whereas it is 1300 $\mu$atm in the M&M, even if target and measurement were not identical, the target should be uniform. Reply: Changes made as suggested.

P6 Line 37 A non significant response may only called a 'trend towards' Reply: Changes made as suggested.

P 7 line 7 the experiment did not last long enough to draw any reliable conclusion on survival rates Reply: As shown by Vaquer-Sunyer & Duarte (2008)'s meta-analysis that 90% of 282 studies experienced LC50 at 4.6 mg O2 L-1 with the mean LC50 for all organisms at 2.1 mg O2 L-1. The median LT50 (460 studies) was 117 hours or nearly 5 days. We therefore still believe that the high survivorship of organisms over 3-6 days in the low O2 treatments indicates a tolerance (acclimation or adaption) to these conditions and is worthy of noting. Nevertheless, we added the phrase "short-term" to the sentence and it reads now "The Baltic species tested were highly resistant to short-term hypoxia and high CO2, alone or in combination, as they experienced very high survival rate across treatments in the relatively short-duration experiment reported here".

P7 line 14 To support this hypothesis you need to add a reference which documents higher mortality for populations from habitats with less abiotic stress Reply: In this particular line (and following), we make a suggestion and already added references.

P7 line 17 and following Please consider that the RI hypothesis and in particular the definition of an exact threshold is still under debate: https://www.biogeosciences.net/10/2815/2013/bg-10-2815-2013.pdf Reply: We have attempted to use Brewer and Peltzers RI's to test its ability to predict marine responses to O2 and CO2 and state in the discussion that we feel it did not hold predictive power in the context of this experiment. We hope this may contribute to the ongoing discussion.

P8 Line 37 Even though calcified structures and the calcification process might be

affected by undersaturation it is not clear why this should be detectable in the rates of aerobic metabolism Reply: We agree and changed the phrasing. It now reads "Hence, the RI does not hold predictive power on the effects of hypoxia and/or pCO2 on the species tested here, which seemed best predicted from consideration of the ranges of O2 and CO2 they experience in their habitat.".

P9 line 6 Hypoxia is necessarily always coupled to elevated CO2 Reply: Changes made as suggested. It now reads "and is, therefore, coupled with elevated pCO2".

Table 3 please give the unit for respiration rates Reply: Changes made as suggested and added the information requested.

Some references are either missing in the text or in the reference list e.g. Grans or Gräns? et al. is not in the list Reply: We checked the reference list and made the changes as suggested.

---

## Author Comment (AC2) · 25 Jan 2018

General comments The topic of the present work is interesting and the authors try to make a correlation between the obtained data and the environmental characteristics of habitats. However, my concern is the period of exposure to particular stressors, which may determine and the range of tolerance of marine invertebrates to environ- mental changes. For example, it is reported (page 8, line 43) that invertebrates from this ecosystem which showed little or no mortality in the presence of both stressors, reflects the range of conditions in the habitats these organisms occupy. However, mortality is depended on several factors including reproduction period, body size, etc. Moreover, it is depended on the period of species exposure to stressors. Also, a key point for the species to withstand stressful conditions for long term is their ability to keep a stable

energy turnover since. Such metabolic responses and patterns determine and their thermal limits. Even more some species live at the edges of their range of thermal tolerance. Thus, long-term experiments might help further not only in estimating species' ability to withstand stressful conditions but to make a better correlation with the future climate projects. I consider that the authors should take into their consideration the above points and to reconsider the interpretation of some obtained data. I agree with author's statement about the complexity of stressors and the challenge of predicting how global stressors will affect marine ecosystems in the future. Reply: We agree with the comments in regards to longer-term experiments, however it is beyond the scope of this work. Our research questions were targeted at being able to detect responses to short-term and acute environmental changes that may occur suddenly as part of eutrophication events. We chose future climate change targets for O2 and CO2 levels as they are realistic representations of what will come in the future, or in some cases might already occur nowadays.

Specific comments Page 2, line 8. This metabolic control. . .could be changed to The involvement of metabolic processes in the regulation of the pH in coastal water is... . .. Reply: Changes made as suggested. It now reads "The involvement of metabolic processes in the regulation of pH in coastal waters is particularly evident when eutrophication stimulates algal blooms, leading to increased sedimentation of organic matter, subsequently degraded through microbial respiration, thereby consuming O2 and releasing CO2 (Conley et al., 2009)".

Page 2, line 15. . . ...although the combined stress from depleted O2 and high CO2 is likely to provide a significant challenge to coastal invertebrates and less mobile species... could be changed to although the synergistic effect of O2 depletion and CO2 accumulation is likely to provide a significant challenge to coastal invertebrates and mostly to sessile species. Reply: Changes made as suggested. It now reads "Whereas scientists have acknowledged this coupling over decades, the impacts of these two pressures have continued to be studied in isolation, although the synergistic

effect of O2 depletion and CO2 accumulation is likely to provide a significant challenge to coastal invertebrates and mostly to sessile species.".

Page 2, lines 24-26. There are many invertebrates tolerant to hypoxia (e.g. mussels). Thus, the authors should be focused on these species which rather are less tolerant (e.g. benthic invertebrates). Reply: As the aim of the study was to test the combination of two stressors, we also used "tolerant" species to see how they react to the combination of the two stressors hypoxia and elevated pCO2. Moreover, it has been shown that responses are highly species specific and not taxa-related (see Fabry, 2008; Malakoff, 2012; Calosi et al., 2013).

Page 2, line 37. I would prefer synergistic instead joint Reply: We made the change as suggested.

Page 2, line 38. ..future levels of what I consider that the two last paragraphs should be reorgasinized and rewritten in such a way so the firstly the authors to be reported at several hypotheses and secondly at their aims Reply: We reorganized the paragraphs as suggested.

Methods and Materials 1. Merge the two first paragraphs Reply: Changes made as suggested.

2. Make clear, when saying history, whether the reported environmental characteristics are long lasting. It is very important since species experiencing such environmental changes in their life cycle may have adapted to such environmental changes by developing the corresponding cellular and physiological mechanisms. Reply: This has been re-worded to show that we are referring to natural and sustained seasonal events which occur in winter and can be exacerbated by nutrient enrichment. It reads now "Both natural and anthropogenically enhanced hypoxia occur within the fjord when enrichment is high and seasonal water exchange over the sill is slow (Josefson and Widbom, 1988; Arneborg, 2004)".

3. Report which of the examined rocky species are exposed or not to air because of tide. The latter characterizes sessile species tolerant to hypoxia. Reply: There is no real tides in the fjord where we collected the speciements. The seawater level can change by a few dm (less than 1 meter) depending on atmospheric pressure, winds, etc. Among the tested species, only Littorina sp. and Mytilus sp. can be occasionally be exposed to air.

4. Change Metabolic response to Metabolic rate or Oxygen consumption. Metabolic responses usually is referred when we examine the metabolic patterns (e.g. enzyme activities, metabolites etc) Reply: Changes made as suggested.

5. Page 5, line 8. For the readers describe briefly the physiological meaning of term respiration index. Reply: We feel that this has been described in the following sentences. But if the editor wants us to describe it in a different way we will add a description.

6. Page 5, lines 15-21. The experimental procedure for determining the oxygen consumption should be written in details. For example, chamber volume, was it the same for all species examined? Reply: We updated the formula in the manuscript to mg L-1 O2 min-1 L-1 g DW-1 as the volume of the glass chamber was included in the calculation. Thus, we don't feel the need to report all the chamber sizes in the manuscript. But if the editor is of the opinion that those data (mean SE for each treatment) are essential for the manuscript, we will off course add this information.

Also report the temperature, salinity and pH of water. It is very important to report the period (hours) of experimental procedure since under a particular level of PO2 metabolism sifts from aerobic to anaerobic and this point is species depended. Reply: The water for incubation had the same values as the experimental aquaria and held in the same room so the temperature and salinity would be the same as reported in Table 2. Incubations lasted a maximum of 5.5 hours, to make sure there is some oxygen left in the glass chamber. None of them reached 0.0 mg L-1 oxygen. We added the max.

incubation time to the manuscript but don't see the need of reporting time (minutes) in detail. But if the editor is the opinion that those data (mean SE for each treatment) are essential for the manuscript, we will off course add this information.

7. Page 5, line 23. Ratio of what? Reply: Changes made as suggested and it reads now "The response ratio of the respiration rate. . .".

Results 1. Respiration. Report the consumption of oxygen rate for each examined species and give possible differences between each other. Reply: The data are provided in Table 3 (mean SE).

2. Give more information the differences or not for the oxygen consumption for each species at each treatment Reply: The data and results of statistical tests and GLM are shown in Table 3. We don't feel the need to report all of them twice and mention them again in the text of the manuscript. But if the editor is of the opinion that it is 100% necessary we will make the change as suggested.

3. In general the results should be rewritten in such a way so to be more clear what is happening in each species at tested treatments and whether differences were recorded from species to species. Reply: Our research questions were more targeted towards the differences between treatments for each species and how their habitats may have played a role.

Table 3. In the column day it is marked 3/6, 3/5 etc. In the legend it is reported pooled data where we had 3 and 6 days. Thus the number 4, 5 2 what do they mean. Reply: These were days of measurement, that were pooled. We have altered the wording to reflect this (in Table 3 and methods).

Discussion Page 7, line 29-30. It is unclear what the authors report. Reply: Changes made for clarity. It now reads "The community in this area has already been sieved of species vulnerable to low O2 concentrations due to a history of hypoxia and even complete anoxia within the last four decades (Nordberg et al., 2000; Polovodova et al.,

2011).".

Page 7, line 32-33. It is very important to report whether such changes in pH regard fluctuations or permanent changes. In the first case the organisms face waves of such changes and how long such waves last. Reply: These just represent fluctuations. This sentence has been reworded to reflect this difference.

Page 7, line 36-37. Community of what? Rewrite the sentence (line 37-39), since it is unclear what it is meaning. Reply: Changes made as suggested. It now reads "Exposed A. filiformis live in sediment burrows that experience much lower oxygen and higher pCO2 than surrounding water which intensifies with depth (Hu et al., 2014). A. filiformis have been shown to withstand a pH of 7.0 and O2 levels below 2.0 mg L-1 and experience no mortality (Hu et al., 2014). Hence, the species tested here already has O2 and pH values comparable to those used as treatments here, particularly for infauna, such as A. filiformis and B. lyrifera which appear to be exposed to low O2 and pH conditions on a regular basis.".

Page 9, line 16. Responses . . .which responses? Reply: Changes made as suggested, it reads now "Respiratory responses".

Page 9, line 1-2. Do you know how long these events last? Is it an acute environmental change or long-term change? Reply: There is an overall trend towards decreasing oxygen over time in the Fjord (based on foraminifera populations) however hypoxic events can vary in duration. There is also a seasonal trend of decreasing oxygen over winter months before new water comes into the fjord.

Page 9, line 10. It could be nice if the authors could support such adaptive responses, genetically determined, by reporting differences from individuals of the same species but from different populations habiting environments differing in the tested abiotic factors. The observed responses in the present work may regard phenotypic plasticity which may be observed and in individuals from populations living in other environments when treated similarly. Reply: We were limited in time and logistics, thus there

was no option for us to test different populations of the same species. But we agree that this should be taken in consideration for future experiments to compare if there are differences in responses depending on populations, water quality and conditions the individuals experienced previously.

———————————————————

---

## Author Response (AR2)

Page 3,at line1 impose and not imposes

Reply: Change made as suggested.

Page 3, line 12, I suspect that it is sea stars and not sea starts.

Reply: Change made as suggested.

Page 5, line 33: Please clarify the units of the respiration rate.

Reply: Change made as suggested, it now reads "DO mg L$^{-1}$ min$^{-1}$ g DW$^{-1}$".

Table 3: I guess that the units of the respiration rate should be mg O2 /l/min /gDW as in Figure 1. Please clarify. and "Of" anf "not off".

Reply: Changes made as suggested.

Figure 3: legend: shown and not show

Reply: Change made as suggested.